# Comprehensive evolutionary analysis and nomenclature of plant G3BPs

Aala A Abulfaraj[1],*, Hajime Ohyanagi[2,3],*, Kosuke Goto[2,4], Katsuhiko Mineta[2], Takashi Gojobori[2], Heribert Hirt[5,6,7], Naganand Rayapuram[5]

**Stress induces extensive reprogramming of mRNA metabolism, which includes the transcription and translation of stress-related genes and the formation of stress granules. RasGAP SH3 domain–binding proteins (G3BPs, also called Rasputins) form a highly conserved family of proteins found throughout eukaryotic evolution, which coordinate signal transduction and posttranscriptional gene regulation and play a key role in the formation of stress granules. G3BPs play a role in osmotic, oxidative, and biotic stress in mammals, and recent results revealed that they play similar functions in higher plants. Although simple eukaryotes such as yeast have only one *G3BP* gene, higher plants show a massive expansion of their *G3BP* genes into distinct subfamilies. However, because this family of genes has not been well-characterized in plants, functions that have evolved during this expansion remain unidentified. Therefore, we carried out a phylogenetic analysis of G3BPs in different eukaryotes, particularly focusing on the green lineage. On the basis of this evolutionary analysis of G3BPs in eukaryotes, we propose a uniform nomenclature for plant G3BPs that should help predict the evolutionary and functional diversification in this family.**

## Introduction

Proteins are made from DNA by rather complex processes involving many regulatory levels. First, DNA undergoes gene transcription forming pre-mRNA, and it undergoes splicing, nuclear export, and a number of posttranscriptional modifications before being translated into a protein. Various proteins regulate the stability and degradation of mRNAs and regulate translation initiation, elongation, and termination. These regulatory proteins link signal transduction with RNA and protein metabolism, maintaining cellular homeostasis and promoting cell survival. Various mRNA-binding proteins (RBPs) assemble to form transcript-specific RNA-protein complexes, named messenger ribonucleoprotein particles (mRNPs), and play a crucial role in the life cycle of mRNAs, which leads to proper regulation of gene expression. mRNPs are dynamic structures involved in controlling all features of mRNA metabolism, including nuclear processing, transport, storage, translation, and decay. The conformational plasticity of RBPs and their capacity to interact with distinct targets are at the basis of the huge regulatory potential of mRNPs (Jonas & Izaurralde, 2013; Castello et al, 2016). Posttranscriptional gene regulation is a major factor contributing to the discrepancy between the transcriptome and proteome, which indicates that mRNAs are subjected to several modifications before functional proteins are produced (Shyu & Wilkinson, 2000). Several RBPs interact with a range of signal transduction components that facilitate rapid cellular responses to environmental stimuli to regulate gene expression at the posttranscriptional level. Moreover, multiple signaling pathways regulate the mRNA translational machinery. For instance, phosphorylation may control mRNA translational activity, turnover, decay, or localization (Braun & Young, 2014).

Plant RBPs play critical roles in various processes that involve regulation of posttranscriptional gene expression ranging from development to adaptation to various environmental conditions. According to the TAIR10 Arabidopsis genome annotation, there are more than 200 RBPs in *Arabidopsis* (Marondedze et al, 2016). The RasGAP SH3 domain–binding proteins (G3BPs) are a highly conserved family of RBPs found throughout eukaryotic evolution. These highly homologous proteins coordinate signal transduction and posttranscriptional gene regulation. All G3BPs have four distinct motifs: (1) a nuclear transport factor 2 (NTF2)–like, (2) an acidic and proline-rich region, (3) an RNA recognition motif (RRM), and (4) an arginine and glycine-rich region (RGG). NTF2-like domains at the N-terminus are not only involved in nuclear transport through

[1]Biological Sciences Department, College of Science and Arts, King Abdulaziz University, Rabigh, Saudi Arabia   [2]Computational Bioscience Research Center, King Abdullah University of Science and Technology (KAUST), Thuwal, Saudi Arabia   [3]Joint Center for Researchers, Associates and Clinicians Data Center, National Center for Global Health and Medicine1-21-1, Tokyo, Japan   [4]Marine Open Innovation (MaOI) Institute Shimizu-Marine Building, Shizuoka, Japan   [5]Center for Desert Agriculture (CDA), King Abdullah University of Science and Technology (KAUST), Thuwal, Saudi Arabia   [6]Institute of Plant Sciences Paris-Saclay IPS2, Centre National de la Recherche Scientifique, Institut National de la Recherche Agronomique, Université Paris-Sud, Université Evry, Université Paris-Saclay, Orsay, France   [7]Max Perutz Laboratories, University of Vienna, Vienna, Austria

Correspondence: naganand.rayapuram@kaust.edu.sa
*Aala A Abulfaraj and Hajime Ohyanagi contributed equally to this work.

nuclear pores but also mediate protein–protein interactions forming homodimers and oligomers (Alam & Kennedy, 2019; Reuper et al, 2021a). Moreover, NTF2 regions can interact with Ran at the nuclear pore, but this remains to be confirmed (Macara, 2001). The acid-rich motifs in the central regions of G3BPs are involved in protein–protein interactions. The proline-rich regions, which are recognized by PxxP motifs, are also found in the central regions of G3BPs. PxxP is the minimal consensus target site that binds to the aromatic amino acids in target SH3 domains (Alam & Kennedy, 2019). The C-termini of G3BPs contains RNA recognition motifs (RRMs) that are involved in RNA binding and have two conserved sequences, RNP1 and RNP2, which interact with RNA through a β-sheet binding platform, and the structural integrity is provided by the α helices (Nagai et al, 1995; Irvine et al, 2004). RGG boxes at the C-termini of G3BPs contain arginine-glycine-glycine and are found in RNA-binding proteins to facilitate RNA-binding, nuclear translocation, and posttranscriptional modifications (Nichols et al, 2000; Abulfaraj et al, 2018; Alam & Kennedy, 2019).

The functional roles of G3BPs in multiple cell signaling pathways across various organisms and under different physiological conditions have been recently reviewed (Alam & Kennedy, 2019). Unlike other organisms, G3BPs in plants have not been characterized yet.

Currently, in the field of plant G3BPs, there is a discrepancy in the naming convention used by different research groups (Krapp et al, 2017; Abulfaraj et al, 2018; Reuper et al, 2021a, 2021b). G3BPs are defined by the presence of at least one NTF2 domain and an RRM domain. Recently, we scanned the entire *Arabidopsis* genome for proteins that satisfy this criteria and identified eight G3BPs (Abulfaraj et al, 2018). However, later, Reuper et al (2021a), restricted the number of G3BPs by considering only those proteins that contained a single NTF2 domain and a single RRM domain (Reuper et al, 2021a). In this article, we focus on the comprehensive phylogenetic analysis of plant G3BPs and propose a nomenclature for Arabidopsis G3BPs. Having a unified classification and nomenclature for naming members of the plant G3BP family will help research on this topic.

## Results

### A comprehensive view of G3BP evolution in eukaryotes

With the aim of comprehensively exploring the genetic divergence in G3BPs throughout eukaryotic evolution, we exhaustively

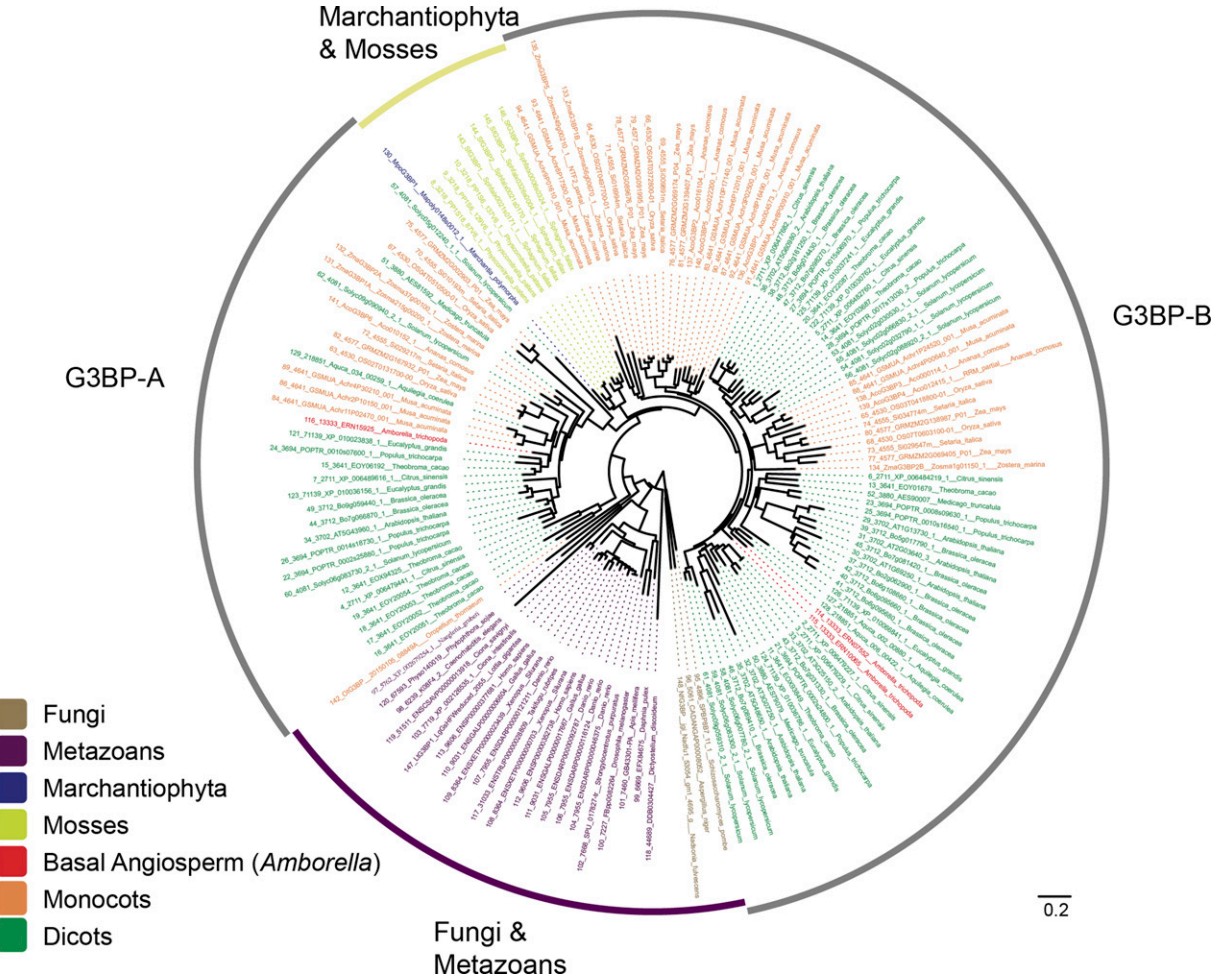

**Figure 1. Phylogenetic tree of 148 G3BP proteins from 39 eukaryotic species (fungi, metazoans, and plants).**
Source data are available for this figure.

collected the eukaryotic G3BP orthologs by using a keyword search of KOG0116 (the eukaryotic cluster of orthologous genes for *G3BPs*) in the EggNOG database. All the proteins were searched against the Pfam-A protein domain structure database by HMMER to ensure that they have both NTF2 and RRM domains (see the Materials and Methods section).

The phylogenetic tree of 922 G3BPs (the full set) is shown in Fig S1 and that of 148 G3BPs (the subset) in Fig 1. Each of them similarly shows that G3BPs of fungi and metazoans are segregated from and distantly related to plant G3BPs. In addition, when the tree is rooted by the fungi-metazoans lineage, the land plant G3BPs are split into two clusters, away from the Marchantiophyta and mosses lineage (Fig 1). It is also observed that each cluster consists of monocots and dicots, as well as *Amborella* (basal angiosperm) orthologs (Fig 1).

### The origin of coexistent NTF2 and RRM domains

In this work, we have focused on eukaryotic G3BPs, with the criteria that they should possess both NTF2 and RRM domains as a prerequisite to be considered as G3BPs. As shown above, the fungal species already had G3BPs. To understand the very early lineage of G3BPs in terms of molecular evolution, it is crucial to determine the first instance when these domains came into coexistence. With the aim to address the origin of coexistent NTF2 and RRM domains in a single gene, we conducted a preliminary search against the prokaryotic dataset (NCBI Conserved Domain Database, https://www.ncbi.nlm.nih.gov/cdd/). Our analysis showed that there are no genes in the prokaryotic lineage which possess both NTF2 and RRM domains.

### Subfamilies in angiosperm G3BPs

To uniformly define and nomenclate the angiosperm G3BP subfamilies, we further conducted a detailed sequence analysis of plant G3BPs. Fig 2 shows the phylogenetic relationship and domain-level similarity of 21 G3BP sequences from five representative species in land plants (*Arabidopsis thaliana*, *Oryza sativa*, *Amborella trichopoda*, *Physcomitrella patens*, and *Marchantia polymorpha*). In addition, an amino acid–level comparison was conducted, particularly in the NTF2 domain (Fig 3A). It demonstrates that there are three amino acid positions that show subfamily-specific amino acid changes (Fig 3B). The amino acid–level comparison in full-length G3BPs is shown in Fig S2.

Our analysis suggests that there exists subfamily-specific genetic diversity at the amino acid level (Fig 3) but not at the protein domain level (Fig 2). Here, we identified two subfamilies in angiosperm G3BPs, that is, G3BP-A of a smaller G3BP group and G3BP-B of a larger G3BP group (Figs 1–3). Mostly with a few exceptions, G3BP-Bs are relatively closely related to outgroups (Marchantiophyta, mosses, fungi, and metazoans) than are G3BP-As (Fig 1). This might suggest that G3BP-Bs are ancestral forms of plant G3BPs, whereas G3BP-As are relatively recent. Also, the amino acid substitutions to W (tryptophan) and Y (tyrosine) are less likely but found at positions 1 and 3 in G3BP-As, suggesting that this substitution might be affected by (adaptive) selection (Henikoff & Henikoff, 1992).

### Gene number expansion of G3BP in plants

Higher plants show a massive expansion of their *G3BP* genes, whereas those of simple eukaryotes such as fungi have a single *G3BP* ortholog (Wang et al, 2012; Abulfaraj et al, 2021). To clearly show the gene number expansion of *G3BPs* in plants, we precisely counted all *G3BP* orthologs in our representative dataset (148 *G3BPs* from 39 species) (Fig 4). This explicitly shows that fungi, metazoans, and ancient land plants (Marchantiophyta) have a single or very small number of *G3BPs*, whereas later land plants have larger numbers of *G3BP* genes (Fig 4). In particular, monocots and dicots show a clear trend of gene number expansion in their lineages, showing 6.3 and 7.5 *G3BP* genes per species on average, respectively (Fig 4). This trend is observed in both G3BP-A and -B

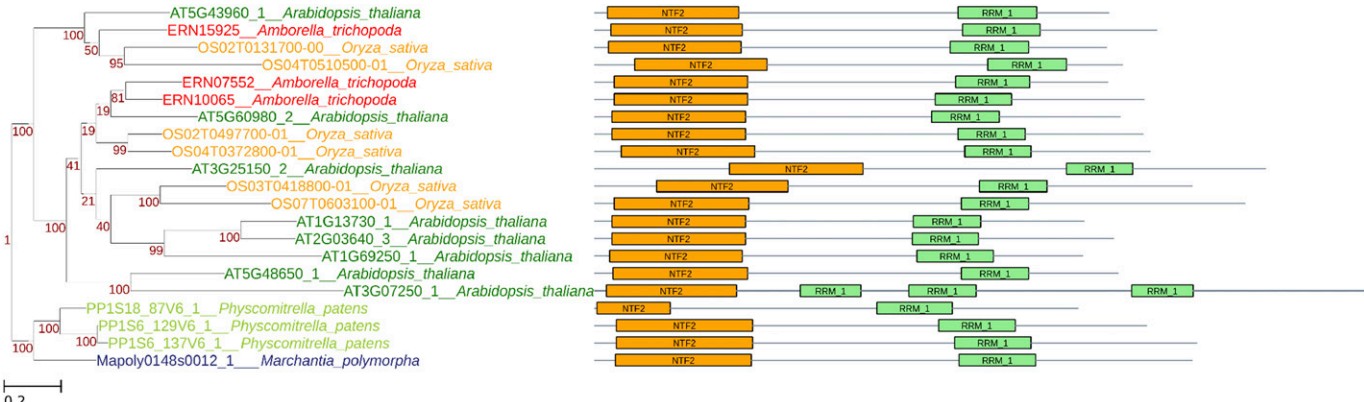

**Figure 2. Phylogenetic relationships in 21 G3BP protein sequences from five representative species in land plants (*Arabidopsis thaliana*, *Oryza sativa*, *Amborella trichopoda*, *Physcomitrella patens*, and *Marchantia polymorpha*).**
A phylogenetic tree of G3BP proteins. Their domain structures detected by searching against Pfam-A database with Hidden Markov Model (hmmscan) were shown on the right (threshold: E-value ≤ 1 × $10^{-4}$ and aligned region ≥ 50% against registered domain length). The positions of G3BP-A–specific amino acid differences detected in the NTF2 domains (AT5G43960, ERN15925, OS02T0131700-0, and OS04T0510500-01) are shown as reversed yellow triangles.
Source data are available for this figure.

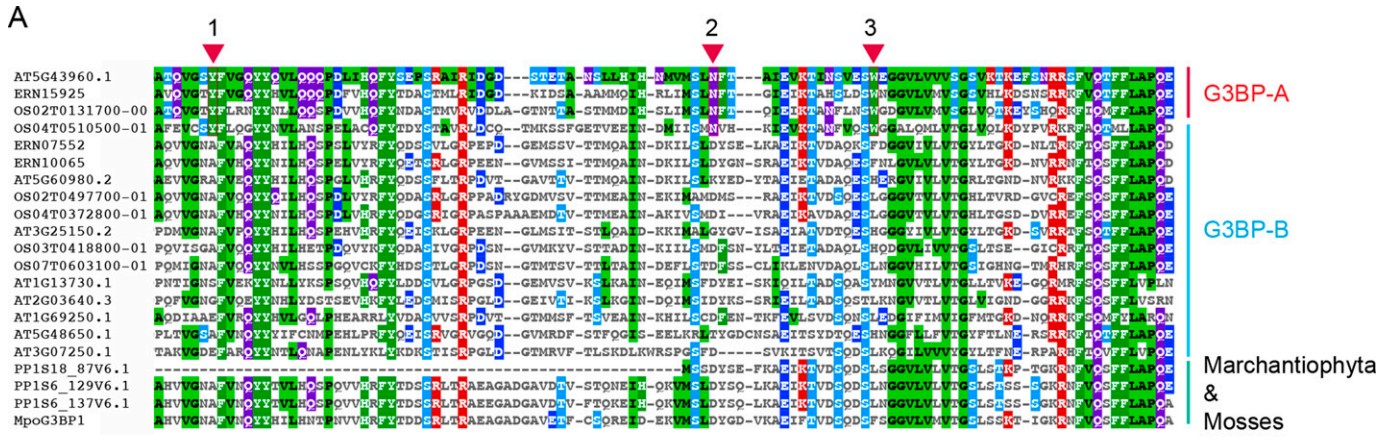

**Figure 3. Aminoacid sequence alignment of the the NTF2 domain in G3BPs.**
**(A)** Alignment of NTF2 domain regions of 21 G3BP protein sequences from five representative species in land plants (*Arabidopsis thaliana*, *Oryza sativa*, *Amborella trichopoda*, *Physcomitrella patens*, and *Marchantia polymorpha*). The three G3BP-A–specific amino acid differences detected are shown in red rectangles. **(B)** Details for the G3BP-A–specific amino acid differences in the four protein sequences. For example, A14Y stands for the consensus A (Alanine) residue at 14[th] position changed to Y (Tyrosine) in G3BP-A.

subfamilies (Fig 4). Comparing G3BP-A and -B subfamilies, the number of expansions is biased toward G3BP-B, showing a contrasting mode of expansions between G3BP-A and -B and explain the different sizes of these subfamilies. As discussed above, the functional differences of these subfamilies might be endorsed by the diversity in the members of the G3BP-B subfamilies. To confirm these findings in an independent manner, we generated a species tree using OrthoFinder ver. 2.5.4 (Emms & Kelly, 2015, 2019) with protein sequences downloaded from JGI (*Oropetium thomaeum*), Ensembl (*Ciona savignyi*), and NCBI (Table S1). And then, an ancestral state of the number of G3BP orthologs was reconstructed by Mesquite ver. 3.70 (Maddison & Maddison, 2021) with a parsimony reconstruction method (Fig S3).

## Discussions

G3BPs form a protein family that is highly conserved during evolution. Given the diverse roles G3BPs play in yeast and mammals, it will be interesting to see functions that have evolved during the expansion of G3BPs in plants. The phylogenetic analysis that we carried out suggests that after the divergence from fungi and metazoans, plants started to increase genetic varieties of G3BPs within their lineage and develop two subfamilies of G3BPs, particularly in the early stage of land plant evolution (i.e., the age of basal angiosperm). It is noteworthy that moss and Marchantiophyta consist of a single clade, indicating the ancestral feature of G3BPs in these species.

Potential confusion about the membership and nomenclature could pose hurdles in the characterization of these important families of proteins. In Viridiplantae, we observe a single gene coding for G3BP among Chlorophytes (green algae) and for *M. polymorpha*, a basal liverwort lineage of land plants. However, on the contrary, other mosses already possess several G3BP copies, and a major independent gene amplification event probably occurred in the common ancestor of angiosperms, thus forming two distinct groups (G3BP-A and G3BP-B). Our phylogenetic analysis clearly shows two distinct modes of evolution in G3BPs. Hence, we designated these G3BP subgroups as G3BP-A and G3BP-B. For each subgroup, we could identify sequence motifs and different evolutionary events such as gene duplications. We suspect that these subgroups serve distinct roles. The genome of the basal angiosperm species *A. trichopoda* contains a single gene coding for one G3BP-A and two genes coding for G3BP-Bs. For most other angiosperms, G3BP-A is still represented by a single member, whereas G3BP-B is largely amplified. Longer branches for certain G3BP-Bs suggest that these genes are under positive selection or relaxation of functional constraints. Functional comparison between these two groups indicates different rates of gene duplication and/or fixation of paralogs. Finally, a huge diversity at the amino acid level of G3BPs occurs among dicots compared with monocots, which is unusual for multigenic families. Based on these results, we propose a systematic nomenclature for the eight *Arabidopsis* G3BPs listed in Table 1 with their new names, group/family to which they belong, Arabidopsis Genome Initiative (AGI) numbers, former name if it already existed, function if known, and the corresponding reference. This nomenclature is the same as the one we had

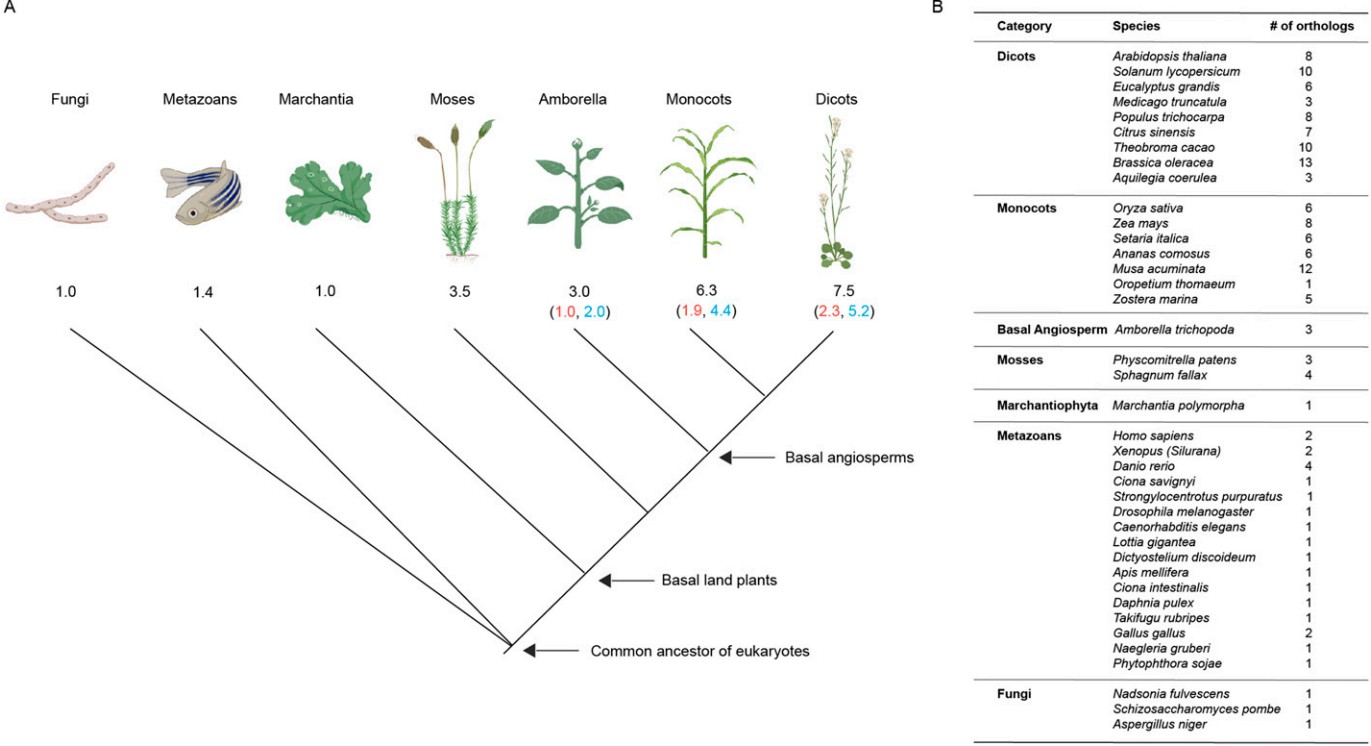

**Figure 4. Change in the number of *G3BP* genes in plant evolutionary history.**
**(A)** A graphical representation of *G3BP* genes during plant evolution. The arrows indicate the common origin of eukaryotic *G3BPs*, the basal land plants, and basal angiosperms. The average number of *G3BPs* in each category is indicated in black. In parenthesis, the average number of G3BP-As is indicated in red, whereas that of G3BP-Bs is indicated in blue. **(B)** Table showing the number of *G3BP* otholog for selected species under different categories.

proposed in an earlier article in 2018 (Abulfaraj et al, 2018). The unified classification and naming of the G3BP superfamily will facilitate connectivity and coherence in future studies on members belonging to this protein family. The gene number expansion of *G3BPs* has exclusively happened in the plant lineage, while plant *G3BP* evolution had been conservative up to the age of basal land plants. Higher land plants, especially in monocots and dicots lineages, started to increase the number of *G3BP* genes in both the G3BP-A and -B subfamilies. This expansion may be caused by the exposure of land plants to critical environmental changes in their biological habitats. Further research on G3BPs should reveal the features of these subgroups to expand and deepen our understanding of the molecular functions of G3BPs.

# Materials and Methods

### G3BP protein sequences

A total of 1,058 G3BPs from 436 eukaryotic species were downloaded from EggNog database version 5.0.0 (Huerta-Cepas et al, 2019), slightly supplemented (19 G3BPs corresponding to different categories such as additional monocots [*Ananas comosus*, *O. thomaeum*, and *Zostera marina*], basal angiosperm [*A. trichopoda*], mosses [*Sphagnum fallax*], *Marchantia* [*M. polymorpha*], metazoans [*Caenorhabditis elegans*], and fungus [*Nadsonia fulvescens*] were added as representatives to cover as wide a gamut of

classification as possible) and corrected by other publicly available protein databases (in total 1,077 G3BPs). Among the 1,077 G3BPs, HMMER search (see below) confirmed that 922 (from 382 eukaryotic species) had at least one each of NTF2 and RRM domains and subjected to the following analyses. The 382 eukaryotic species include fungi, metazoans, *Marchantiophyta*, mosses, an *Amborella* species, monocots, and dicots. As a representative dataset, the G3BP subset (148 G3BPs from 39 species) was selected by manual curation. The 39 species of this subset are listed in Table S1.

### Construction of phylogenetic trees

The multiple protein sequence alignments were generated by MAFFT version7 (Katoh & Standley, 2013) with the option of E-INS-I strategy. Other options for sequence alignment were set to default values. The distance matrices were calculated based on JTT amino acid substitution model, and the phylogenetic trees were constructed by the neighbor-joining method (Saitou & Nei, 1987). The unrooted trees were operationally rooted by the fungi-metazoans lineage. Bootstrap values were calculated by 100 times iterations, in the case of 21 G3BP sequences from five representative species in land plants.

### Protein domain structure detection

The protein domain composition were detected by hmmscan program implemented in HMMER version 3.1b1 (Mistry et al, 2013)

**Table 1. List of *Arabidopsis* G3BPs with novel nomenclature**

| Name | Family | AGI number | Former name | Function | Reference |
|---|---|---|---|---|---|
| AtG3BP7 | A | At5G43960 | AtG3BP-like AtG3BP2 | Virus resistance induced upon virus infection | Krapp et al (2017), Abulfaraj et al (2018), and Reuper et al (2021a, 2021b) |
| AtG3BP6 | | At3G25150 | AtG3BP3 | Induced upon virus infection | Abulfaraj et al (2018) and Reuper et al (2021a, 2021b) |
| AtG3BP5 | | At3G07250 | | Not yet characterized | Abulfaraj et al (2018) |
| AtG3BP1 | | At5G48650 | AtG3BP1 AtG3BP7 | Regulator of stomatal and apoplastic immunity induced upon virus infection | Abulfaraj et al (2018) and Reuper et al (2021a, 2021b) |
| AtG3BP8 | B | At5G60980 | AtG3BP1 | Induced upon virus infection | Abulfaraj et al (2018) and Reuper et al (2021a, 2021b) |
| AtG3BP2 | | At1G13730 | AtG3BP5 | Not induced upon virus infection | Abulfaraj et al (2018) and Reuper et al (2021a, 2021b) |
| AtG3BP4 | | At2G03640 | AtG3BP6 | Induced upon virus infection | Abulfaraj et al (2018) and Reuper et al (2021a, 2021b) |
| AtG3BP3 | | At1G69250 | AtG3BP4 | Induced upon virus infection | Abulfaraj et al (2018) and Reuper et al (2021a, 2021b) |

with Pfam-A profiles of the hidden Markov model (Pfam33.1, released on 2 May 2020) (Mistry et al, 2021).

### Data visualization

Each phylogenetic tree was drawn and colored by FigTree version 1.4.3 (http://tree.bio.ed.ac.uk/software/figtree/). The protein domain structures were coordinately visualized with their phylogenetic relationships (trees) by ETE Toolkit version 3.0 (Huerta-Cepas et al, 2016). The multiple protein sequence alignments were visualized by MView (Brown et al, 1998).

# Supplementary Information

# Acknowledgements

This work was supported by King Abdullah University of Science and Technology (BAS/1/1062-01-01 to Prof. H Hirt and BAS/1/1059-01-01 to Prof. T Gojobori). We would like to acknowledge Prof. Jean-Marc Deragon and Prof. Cecile Bousquet-Antonelli for valuable discussions and suggestions.

## Author Contributions

AA Abulfaraj: formal analysis, methodology, and writing—original draft, review, and editing.
H Ohyanagi: formal analysis, validation, investigation, visualization, methodology, and writing—original draft, review, and editing.
K Goto: formal analysis, investigation, methodology, and writing—original draft.
K Mineta: formal analysis, investigation, methodology, and writing—original draft.
T Gojobori: supervision, validation, project administration, and writing—review and editing.
H Hirt: conceptualization, supervision, validation, project administration, and writing—review and editing.
N Rayapuram: conceptualization, data curation, formal analysis, supervision, methodology, project administration, and writing—original draft, review, and editing.

## Conflict of Interest Statement

The authors declare that the research was conducted in the absence of any commercial or financial relationships that could be construed as a potential conflict of interest.

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
