## [Reviewer comments · Life Science Alliance]

Life Science Alliance

Comprehensive Evolutionary Analysis and Nomenclature of Plant G3BPs

Aala Abulfaraj, Hajime Ohyanagi, Kosuke Goto, Katsuhiko Mineta, Takashi Gojobori, Heribert Hirt, and Naganand Rayapuram
DOI: <https://doi.org/10.26508/lsa.202101328>

Corresponding author(s): Naganand Rayapuram, King Abdullah University of Science and Technology

Review Timeline:

Submission Date:	2021-12-02
Editorial Decision:	2022-01-25
Revision Received:	2022-03-31
Editorial Decision:	2022-05-05
Revision Received:	2022-05-11
Accepted:	2022-05-12

Scientific Editor: Novella Guidi

Transaction Report:

January 25, 2022

Re: Life Science Alliance manuscript #LSA-2021-01328

Dr. Naganand Rayapuram
King Abdullah University of Science and Technology
Center for Desert Agriculture
4700
Thuwal, Makkah 23955
Saudi Arabia

Dear Dr. Rayapuram,

Thank you for submitting your manuscript entitled "Comprehensive Evolutionary Analysis and Nomenclature of Plant G3BPs" to Life Science Alliance. The manuscript was assessed by expert reviewers, whose comments are appended to this letter. We, thus, encourage you to submit a revised version of the manuscript back to LSA that responds to all of the reviewers' points.

Thank you for this interesting contribution to Life Science Alliance. We are looking forward to receiving your revised manuscript.

Sincerely,

B. MANUSCRIPT ORGANIZATION AND FORMATTING:

Reviewer #1 (Comments to the Authors (Required)):

The authors present, in the submitted manuscript, their results on the analysis of the sequence similarity and phylogeny of the Rasputins, also known as G3BP, a widespread group of eukaryotic proteins involved in mRNA metabolism and characterized by the presence of an NTF-2 domain and, at least, one RRM domain. The objective of this work is of great interest, due to the fact that this protein family is mostly unexplored in plants and the authors identify interesting episodes of gene duplications and lineage specific expansion.

In its current form, unfortunately, the manuscript does not make a real contribution to the literature because (1) the authors have failed to demonstrate that their collection of rasputins and rasputin-related protein is exhaustive, therefore claims that basal lineages have one or few copies will be receive with skepticism; (2) the phylogenetic analysis performed for this manuscript needs further refinement and a more thorough presentation. Since the authors did not show the bootstrap values of most of the clades in their phylogenies, it is not clear which results are expected to be reasonably stable enough to serve as basis for a new nomenclature.

Importantly, more detailed comparisons with reference organismal phylogenies, at different taxonomic levels, is required for rigorous inference of gene gain/loss/duplication events. Since the data is voluminous, the authors may choose to use automated gene/gain/loss inference packages, such as R's ape library, Mesquite or Hyphy.

With regard to the methods used in this study, it is notable that the high quality ETE3 pipeline (eggNOG41) of the ortholog database EggNOG, which is the main data source for this study, is not mentioned but generates a tree that is probably different from the one presented in this manuscript. Unfortunately, I can't verify this because the main phylogeny in this work is just a low resolution image (Suppl. Fig. 2), which is not browsable, searchable or can be used in further analysis. Providing this data is an absolute requirement for both revision and reproducibility.

In conclusion, the current manuscript needs major revisions, with a more detailed and explicit presentation of data and should further explore the distribution of G3BP homologs in all eukaryotic lineages and in the plants. In its current presentation, the manuscript is unfit for publication.

Below, I present suggestions for improvement and both major and minor corrections that need to be done.

- 1) The authors did not verify the relationship of the Rasputins and other NTF2-containing proteins
The main problem here is that genome annotation might generate a swarm of artefactual fragmented genes in non-model organisms, thus leading to false negatives. Also, it is important to verify whether pseudogenes of this family are present in any lineages, which would lead to better understanding of the gain and loss events. Searches on conceptual translations of the target genomic sequences is capable of verifying whether such pseudogenes exist;
- 2) Iterative searches of individual domains should be performed against the target proteomes to make sure all NTF2+RRM proteins are collected and none is lost due to sequence divergence beyond the detection level of Pfam models;
- 3) Removal of poorly aligned columns is a common strategy to reduce uncertainty in phylogenetic inference and is used in EggNOG's phylogenetic pipeline. The authors should use a similar strategy or present a clear justification for not doing so.
- 4) In page 4, the authors say 19 sequences were added and corrected but they do not say which sequences, how these sequences were chosen and under what criteria;
- 5) In page 5, replace "domain structures" with "protein domain composition" to avoid confusion with the recent state-of-the-art 3D structure prediction methods;
- 6) In page 6, replace "intimate molecular-level analysis" with "detailed sequence analysis"
- 7) In page 8, replace "modes of evolution" with "different rates of gene duplication and/or fixation of paralogs"

8) The authors claim that there are no variations in domain composition among the analyzed proteins but it is important to note that their method of collection of sequences, filtering members of a single EggNOG cluster based on matches to Pfam domains, is less sensitive than direct, iterative, sequence searches against the same proteome databases. Such searches often reveal episodes of domain accretion that might have been overlooked in the current version of this manuscript. Additionally, episodes of internal duplication of the RRM domains are easy to find in Pfam and in EggNOG, but the authors did not mention this detail.

Reviewer #2 (Comments to the Authors (Required)):

The manuscript "Comprehensive Evolutionary Analysis and Nomenclature of Plant G3BPs" by Abulfaraf and coworkers analyzes, by bioinformatic tools, the phylogeny of different G3BPs in eukaryotes, focusing on the G3BPs found in plants. The authors propose to standardize a nomenclature for these proteins in plants.

General comments:

- I found this work of high relevance and I believe it may be improved considering:
- Other groups have used different names to identify G3BP's plants in the literature. The unification of these names would significantly contribute to the study of their functions in each of the species.
- I suggest the authors explain in the text why and how the discrepancy about G3BP naming with the Reuper et al. is generated (<https://doi.org/10.1038/s41598-021-81276-7>; <https://doi.org/10.3390/ijms22126287>)
- In addition, if possible, I suggest adding in panel b of figure 4 if these orthologs of AtG3BPs could coincide with orthologs of other species (e.g.: name system of ribosomal proteins <https://doi.org/10.1016/j.sbi.2014.01.002>).

Minor comments:

- I suggest explaining in the legend of Figure4, panel A what do each arrow and the blue/red numbers between parentheses mean?
- I suggest adding references in Table 1 for each of the AtG3BPs and a column with the nomenclature used in the other studies (eg: <https://doi.org/10.1038/s41598-021-81276-7>; <https://doi.org/10.3390/biom10040661>)
- I suggest fixing the references' format to improve the data search

Reviewer #1 (Comments to the Authors (Required)):

The authors present, in the submitted manuscript, their results on the analysis of the sequence similarity and phylogeny of the Rasputins, also known as G3BP, a widespread group of eukaryotic proteins involved in mRNA metabolism and characterized by the presence of an NTF-2 domain and, at least, one RRM domain. The objective of this work is of great interest, due to the fact that this protein family is mostly unexplored in plants and the authors identify interesting episodes of gene duplications and lineage specific expansion.

In its current form, unfortunately, the manuscript does not make a real contribution to the literature because (1) the authors have failed to demonstrate that their collection of rasputins and rasputin-related protein is exhaustive, therefore claims that basal lineages have one or few copies will be received with skepticism; (2) the phylogenetic analysis performed for this manuscript needs further refinement and a more thorough presentation. Since the authors did not show the bootstrap values of most of the clades in their phylogenies, it is not clear which results are expected to be reasonably stable enough to serve as basis for a new nomenclature.

In order to demonstrate that none of the Rasputin or Rasputin-related proteins are lost due to sequence diversity in our analysis, as a test case, we chose *Shizosaccharomyces pombe* (PomBase, <https://www.pombase.org>), one of the well-annotated and distantly-related species from *A. thaliana* in our study, we ran a comprehensive BLAST search using the query sequences of all eight *A. thaliana* G3BP proteins (AT1G13730.1, AT1G69250.1, AT2G03640.3, AT3G07250.1, AT3G25150.2, AT5G43960.1, AT5G48650.1, and AT5G60980.2) against the *S. pombe* proteome (BLASTP) and genome (TBLASTN) databases. In this additional analysis, we did not find any more G3BP-like proteins with both NTF2 and RRM domains using the 1e-4 threshold. This analysis confirms that we have not missed any potential Rasputins or Rasputin-related proteins (G3BPs).

In terms of phylogenetic stability, our current version of Fig.2 demonstrates the phylogenetic tree of the green lineage represents solid bootstrap results (100 times iterations). For your convenience, we have now included a second version of Fig. 1 (Fig-1-for-reference, see below) that includes the bootstrap values (100 times iterations). Because the proteins are lowly-diversified in gap-free region that we employed for the phylogeny reconstruction (see our reply to your comment #3), several branching sites are unstable and have low bootstrap values. However, the point is that, the phylogeny of Fig-1-for-reference is consistent with our revised *Arabidopsis* G3BP-A and G3BP-B nomenclature (Fig-1-for-reference). In this sense, we could conclude that the phylogeny is stable in terms of G3BP-A/B nomenclature.

Importantly, more detailed comparisons with reference organismal phylogenies, at different taxonomic levels, is required for rigorous inference of gene gain/loss/duplication events. Since the data is voluminous, the authors may choose to use automated gene/gain/loss inference packages, such as R's ape library, Mesquite or Hyphy.

Following the reviewer's suggestion, we have now generated a species tree that was constructed by OrthoFinder ver. 2.5.4 with protein sequences downloaded from JGI (*Oropetium thomaeum*), Ensembl (*Ciona savignyi*), and NCBI (Supplemental Table 1). And then, an ancestral state of the number of G3BP orthologs was reconstructed by

Mesquite ver. 3.70 with a parsimony reconstruction method. We have included the output of this analysis as Supplementary figure 3.

Supplemental Table 1. Protein sequences of 39 representative species.

category	species	proteins
Dicots	Arabidopsis thaliana	GCF_000001735.4
	Solanum lycopersicum	GCF_000188115.4
	Eucalyptus grandis	GCF_016545825.1
	Medicago truncatula	GCF_003473485.1
	Populus trichocarpa	GCF_000002775.4
	Citrus sinensis	GCF_000317415.1
	Theobroma cacao	GCF_000208745.1
	Brassica oleracea	GCF_000695525.1
	Aquilegia coerulea	GCA_002738505.1
Monocots	Oryza sativa	GCF_001433935.1
	Zea mays	GCF_902167145.1
	Setaria italica	GCF_000263155.2
	Ananas comosus	GCF_001540865.1
	Musa acuminata	GCF_000313855.2
	Oropetium thomaeum	Othomaeum_386_v1.0
	Zostera marina	GCA_001185155.1
Basal Angiosperm	Amborella trichopoda	GCF_000471905.2
Mosses	Physcomitrella patens	GCF_000002425.4
	Sphagnum fallax	GCA_021442195.1
Marchantiophyta	Marchantia polymorpha	GCA_003032435.1
Metazoans	Homo sapiens	GCF_000001405.39
	Xenopus tropicalis	GCF_000004195.4
	Danio rerio	GCF_000002035.6
	Ciona savignyi	Ciona_savignyi.CSAV2.0
	Strongylocentrotus purpuratus	GCF_000002235.5
	Drosophila melanogaster	GCF_000001215.4
	Caenorhabditis elegans	GCF_000002985.6
	Lottia gigantea	GCF_000327385.1
	Dictyostelium discoideum	GCF_000004695.1
	Apis mellifera	GCF_003254395.2
	Ciona intestinalis	GCF_000224145.3
	Daphnia pulex	GCF_021134715.1
	Takifugu rubripes	GCF_901000725.2
	Gallus gallus	GCF_016699485.2
	Naegleria gruberi	GCF_000004985.1
	Phytophthora sojae	GCF_000149755.1
Fungi	Nadsonia fulvescens	GCA_001661315.1
	Schizosaccharomyces pombe	GCF_000002945.1

Figure S4: Ancestral state of the number of G3BP orthologs among 39 representative species. Colors show the number of orthologs. Left and right semicircles represent the minimum and the maximum numbers in reconstruction.

With regard to the methods used in this study, it is notable that the high quality ETE3 pipeline (eggnog41) of the ortholog database EggNOG, which is the main data source for this study, is not mentioned but generates a tree that is probably different from the one presented in this manuscript. Unfortunately, I can't verify this because the main phylogeny in this work is just a low resolution image (Suppl. Fig. 2), which is not browsable, searchable or can be used in further analysis. Providing this data is an absolute requirement for both revision and reproducibility.

The EggNOG output is not used for graphical representation in the manuscript as the format is not suitable for publication due to the vast number of proteins in the analysis. Therefore, we reconstructed the phylogeny with JTT amino acid substitution model and Neighbor-Joining method, followed by the tree visualization with FigTree software (see Materials and Methods for details) so that it can be easily comprehensible for the readers. The submission process did not permit us to upload high-resolution images due to size limitations. We have vastly improved the resolution of the images and if necessary, we have now contacted the editorial office to find a way to submit images of high-resolution.

In conclusion, the current manuscript needs major revisions, with a more detailed and explicit presentation of data and should further explore the distribution of G3BP homologs in all eukaryotic lineages and in the plants. In its current presentation, the manuscript is unfit for publication.

Below, I present suggestions for improvement and both major and minor corrections that need to be done.

1) The authors did not verify the relationship of the Rasputins and other NTF2-containing proteins. The main problem here is that genome annotation might generate a swarm of artefactual fragmented genes in non-model organisms, thus leading to false negatives. Also, it is important to verify whether pseudogenes of this family are present in any lineages, which would lead to better understanding of the gain and loss events. Searches on conceptual translations of the target genomic sequences is capable of verifying whether such pseudogenes exist;

Thank you for the insightful comment on false negatives (due to poor genome quality in non-model organisms) and pseudogenes.

In Figure 4, we would like to review a couple of examples. We have the numbers of orthologs in each species (Figure 4B) and the average number of orthologs in each lineage (Figure 4A). When we focused on dicots, *A. thaliana* (8 orthologs, a model organism) and all dicots including both model and non-model organisms (7.5 orthologs on average) show much the same numbers. We did not see clear decrease in number of orthologs in non-model species. The same tendency has been observed in monocot species (6 orthologs for *O. sativa* - a model organism, while 6.3 orthologs for average monocots). We understand that there are some exceptional species (e.g. *Musa acuminata*=12, *Oropetium thomaeum*=1), while we operationally defined the G3BP orthologs referring to EggNOG database. We believe that our discussion is sound at the lineage level.

For the pseudogene issue, as we stated in **Introduction**, G3BPs are defined by the presence of at least one NTF2 domain and one RRM domain. In that sense, pseudogenes are out of scope in this project, but we consider that they are key factors for further investigation for the precise evolutionary process of plant G3BPs. This could be our next target in future analysis.

2) Iterative searches of individual domains should be performed against the target proteomes to make sure all NTF2+RRM proteins are collected and none is lost due to sequence divergence beyond the detection level of Pfam models;

Thank you for the comment. In order to demonstrate that none is lost due to sequence diversity in our analysis, as a test case, we picked up *Shizosaccharomyces pombe* (PomBase, <https://www.pombase.org>), one of the well-annotated and most distantly-

related species from *A. thaliana* in this study. With the query sequences of all the eight *A. thaliana* G3BP proteins (AT1G13730.1, AT1G69250.1, AT2G03640.3, AT3G07250.1, AT3G25150.2, AT5G43960.1, AT5G48650.1, and AT5G60980.2), we conducted exhaustive BLAST search against not only *S. pombe* proteome (BLASTP) database, but also genome (TBLASTN) database. With the threshold of $1e^{-4}$, we did not detect any more G3BP-like proteins with both of NTF2 and RRM domains.

3) Removal of poorly aligned columns is a common strategy to reduce uncertainty in phylogenetic inference and is used in EggNOG's phylogenetic pipeline. The authors should use a similar strategy or present a clear justification for not doing so.

We intentionally included poorly aligned sequences as this allows to demonstrate phylogenetic inference in alignment level (Fig.3). In our actual phylogeny reconstruction step of Neighbor-Joining method, we exclusively employed all the gap-free sites in the alignment.

4) In page 4, the authors say 19 sequences were added and corrected but they do not say which sequences, how these sequences were chosen and under what criteria;

The 19 sequences corresponding to different categories such as additional monocots (*Ananas comosus*, *Oropetium thomaeum* and *Zostera marina*), Basal Angiosperm (*Amborella trichopoda*), Mosses (*Sphagnum fallax*), Marchantia (*Marchantia polymorpha*), Metazoans (*Caenorhabditis elegans*) and fungus (*Nadsonia fulvescens*) were added as representatives to cover as wide a gamut of classification as possible.

We have now added this information to the materials and methods part of the manuscript.

5) In page 5, replace "domain structures" with "protein domain composition" to avoid confusion with the recent state-of-the-art 3D structure prediction methods;

We have now made the change as suggested by the reviewer.

6) In page 6, replace "intimate molecular-level analysis" with "detailed sequence analysis"

We have now brought about this modification.

7) In page 8, replace "modes of evolution" with "different rates of gene duplication and/or fixation of paralogs"

We have now replaced the phrase with that suggested by the reviewer.

8) The authors claim that there are no variations in domain composition among the analyzed proteins but it is important to note that their method of collection of sequences, filtering members of a single EggNOG cluster based on matches to Pfam domains, is

less sensitive than direct, iterative, sequence searches against the same proteome databases. Such searches often reveal episodes of domain accretion that might have been overlooked in the current version of this manuscript. Additionally, episodes of internal duplication of the RRM domains are easy to find in Pfam and in EggNOG, but the authors did not mention this detail.

Thank you for your comment. Each G3BP must have at least one NTF2 and RRM domain, according to the definition (please see Introduction). As a result, we always accepted G3BPs with duplicated NTF2 or RRM domains, which produced the results stated in the paper. In this sense, we are interested in the existence of both NTF2 and RRM domains, but we are not interested in the variation in domain composition (number) in this work, even if we did find a variation in the number of RRM domains in one of the Arabidopsis G3BPs (AT3G07250.1).

Reviewer #2 (Comments to the Authors (Required)):

The manuscript "Comprehensive Evolutionary Analysis and Nomenclature of Plant G3BPs" by Abulfaraj and coworkers analyzes, by bioinformatic tools, the phylogeny of different G3BPs in eukaryotes, focusing on the G3BPs found in plants. The authors propose to standardize a nomenclature for these proteins in plants.

General comments:

- I found this work of high relevance and I believe it may be improved considering:

- Other groups have used different names to identify G3BP`s plants in the literature. The unification of these names would significantly contribute to the study of their functions in each of the species.

- I suggest the authors explain in the text why and how the discrepancy about G3BP naming with the Reuper et al. is generated (<https://doi.org/10.1038/s41598-021-81276-7>; <https://doi.org/10.3390/ijms22126287>)

By definition, G3BPs are proteins that contain at least one NTF2 and one RRM domain. In our first publication, Abulfaraj et al., 2018 in Life Science Alliance, we clearly identified these 8 proteins from the Arabidopsis genome and after generating a phylogenetic tree with the Human G3BP, we attributed names to all of the proteins. We mentioned that we need to unify the nomenclature for the sake of the community. However, Reuper et al., 2021 in Scientific reports, restricted their list of G3BPs by considering only those proteins that contained a single NTF2 and RRM domain, therefore they ended up with only 7 G3BPs. Further, as we have mentioned in Table1, they gave different names and it is not clear what the basis of this new naming system is.

To make this point clear, we have now added the following line in the last paragraph of the introduction.

“G3BPs are defined by the presence of at least one NTF2 domain and a RRM domain. Recently, we scanned the entire Arabidopsis genome for proteins that satisfy this criteria and identified 8 G3BPs (Abulfaraj et al 2018). However, later Reuper et al., restricted the number of G3BPs by considering only those proteins that contained a single NTF2 domain and a single RRM domain (Reuper et al 2021a).”

- In addition, if possible, I suggest adding in panel b of figure 4 if these orthologs of AtG3BPs could coincide with orthologs of other species (e.g.: name system of ribosomal proteins <https://doi.org/10.1016/j.sbi.2014.01.002>).

Currently, we detected all the G3BP orthologues in an EggNOG Cluster, i.e., an Orthologue Cluster. We did not conduct one-by-one orthology detection analysis. As you mentioned, the orthologous correspondence among multiple species is of immediate interest to evolutionary scientists. Now we incorporated another visualization (Figure S3) for further investigation of the orthologous relationships among multiple species. We hope this change is in line with your suggestion.

Minor comments:

- I suggest explaining in the legend of Figure4, panel A what do each arrow and the blue/red numbers between parentheses mean?

We have now included the explanation for what the arrows and blue/red numbers means in the Figure legend. We thank the reviewer for helping us improve the readability of the figure.

- I suggest adding references in Table 1 for each of the AtG3BPs and a column with the nomenclature used in the other studies (eg: <https://doi.org/10.1038/s41598-021-81276-7>; <https://doi.org/10.3390/biom10040661>)

We thank the referee for the nice suggestion. We have now added references for all the AtG3BPs along with a column for nomenclature as well as known function so far.

- I suggest fixing the references' format to improve the data search

We have now fixed the issue with the references to improve the data search as suggested by the reviewer.

May 5, 2022

RE: Life Science Alliance Manuscript #LSA-2021-01328R

Dr. Naganand Rayapuram
King Abdullah University of Science and Technology
Center for Desert Agriculture
4700
Thuwal, Makkah 23955
Saudi Arabia

Dear Dr. Rayapuram,

Thank you for submitting your revised manuscript entitled "Comprehensive Evolutionary Analysis and Nomenclature of Plant G3BPs". We would be happy to publish your paper in Life Science Alliance pending final revisions necessary to meet our formatting guidelines.

- please provide the the browsable raw data files in newick or nexus format of the different phylogenies and include them in the Supplementary Material as requested by Reviewer 1
- please consult our manuscript preparation guidelines <https://www.life-science-alliance.org/manuscript-prep> and make sure your manuscript sections are in the correct order;
- please separate the Results and Discussion section into two - 1. Results 2. Discussion, as per our formatting requirements
- please double-check your Figure 4 legend; the figure has panels A and B, but these are not designated in the figure legend

Figure Issues:

- Figure s1 isn't well readable. Please provide higher resolution.

A. FINAL FILES:

B. MANUSCRIPT ORGANIZATION AND FORMATTING:

Sincerely,

Reviewer #1 (Comments to the Authors (Required)):

The authors have resubmitted an improved version of their former manuscript on the evolution of G3BP protein family, a.k.a rasputins. Once again, it is clear that the most important finding is the division of G3BP's in two main subgroups.

The authors did accept most minor suggestions from both reviewers, making changes that actually improved readability and also added new analysis based that implemented some of my suggestions.

I believe the issue of making sure all G3BP homologs were detected was approached in a satisfying way.

Unfortunately, though, the authors did not understand that when asked to provide browsable phylogenies I was not referring to images, but to the raw data files, in newick or nexus format. These files are the ones that should be included in the Supplemental Material, not just larger images. Actually, these raw data files will be much easier to upload and I strongly suggest the authors and the journal make these files available.

All things considered, I think the article is ready for publication.

May 12, 2022

RE: Life Science Alliance Manuscript #LSA-2021-01328RR

Dr. Naganand Rayapuram
King Abdullah University of Science and Technology
Center for Desert Agriculture
4700
Thuwal, Makkah 23955
Saudi Arabia

Dear Dr. Rayapuram,

Thank you for submitting your Research Article entitled "Comprehensive Evolutionary Analysis and Nomenclature of Plant G3BPs". It is a pleasure to let you know that your manuscript is now accepted for publication in Life Science Alliance. Congratulations on this interesting work.

DISTRIBUTION OF MATERIALS:

Again, congratulations on a very nice paper. I hope you found the review process to be constructive and are pleased with how the manuscript was handled editorially. We look forward to future exciting submissions from your lab.

Sincerely,
